# Morphological Diversity of Dps Complex with Genomic DNA

**DOI:** 10.3390/ijms24108534

**Published:** 2023-05-10

**Authors:** Yuri Chesnokov, Roman Kamyshinsky, Andrey Mozhaev, Eleonora Shtykova, Alexander Vasiliev, Ivan Orlov, Liubov Dadinova

**Affiliations:** 1Shubnikov Institute of Crystallography of Federal Scientific Research Centre “Crystallography and Photonics”, Russian Academy of Sciences, Leninskiy Prospect, 59, 119333 Moscow, Russia; kamyshinsky.roman@gmail.com (R.K.); a.a.mozhaev@gmail.com (A.M.); eleonora.shtykova@gmail.com (E.S.); a.vasiliev56@gmail.com (A.V.); desueagle@gmail.com (I.O.); lubovmsu@mail.ru (L.D.); 2National Research Center “Kurchatov Institute”, Akademika Kurchatova pl., 1, 123182 Moscow, Russia; 3Shemyakin-Ovchinnikov Institute of Bioorganic Chemistry, Russian Academy of Sciences, Miklukho-Maklaya, 16/10, 117997 Moscow, Russia; 4Moscow Institute of Physics and Technology, Institutsky per. 9, 141701 Dolgoprudny, Russia

**Keywords:** DNA-binding protein from starved cells (Dps), DNA, biocrystallization, co-crystals, cryo-electron tomography

## Abstract

In response to adverse environmental factors, *Escherichia coli* cells actively produce Dps proteins which form ordered complexes (biocrystals) with bacterial DNA to protect the genome. The effect of biocrystallization has been described extensively in the scientific literature; furthermore, to date, the structure of the Dps–DNA complex has been established in detail in vitro using plasmid DNA. In the present work, for the first time, Dps complexes with *E. coli* genomic DNA were studied in vitro using cryo-electron tomography. We demonstrate that genomic DNA forms one-dimensional crystals or filament-like assemblies which transform into weakly ordered complexes with triclinic unit cells, similar to what is observed for plasmid DNA. Changing such environmental factors as pH and KCl and MgCl_2_ concentrations leads to the formation of cylindrical structures.

## 1. Introduction

Unlike eukaryotic cells, for which various DNA-containing structures are compactly located in the nucleus, most non-nuclear prokaryotes, including bacteria, contain a single circular chromosome with a genomic size of up to 5 million base pairs (bp) and a length of about 1–2 mm. In cells, it is compacted into a nucleus-like structure called a nucleoid with a tiny volume of ~0.5–3.0 femtoliters. The compaction of DNA in the nucleoid is achieved with the participation of nucleoid-associated proteins (NAPs) [1,2]. NAPs also play essential roles in processes associated with DNA function such as recombination, repair, replication, transcription [3,4], and, importantly, performing DNA protective functions [5].

Therefore, NAPs play a critical role in facilitating a highly efficient and immediate response to numerous stimuli. These proteins are involved in shaping the architecture of bacterial chromosomal DNA in response to both extracellular and intracellular factors. In the presence of significant stress signals, for example, NAPs such as HU and Dps can coat or consolidate the nucleoid, thereby creating a physical barrier for DNA protection [6,7,8,9,10]. Moreover, NAPs such as H-NS and Fis can regulate gene transcription through the structural and topological alterations of DNA induced upon binding to certain sites [11,12]. This in turn leads to changes in the expression levels of particular genes involved in cells’ adaptation to challenging environmental conditions, such as those that drive biofilm formation, changes in motility or virulence mechanisms, and the synthesis of secondary metabolites. Furthermore, NAPs can influence basic cellular processes, such as the regulation of replication initiation, to ensure their coordination with changes in environmental conditions [13,14]. Thus, most NAPs function as rapid reaction forces that help bacteria to endure stress. In general, NAPs have a major impact on nucleoid function as well as bacterial viability and virulence.

These proteins are highly plentiful in the cell, but their abundance depends on the response of the nucleoid to different conditions. For instance, the NAP Dps (DNA-binding protein of starved cells) is particularly abundant in the late stationary phase and becomes the main component of the nucleoid [1,15], sequestering iron [16,17,18] and effectively protecting DNA from external negative influences, including antibiotics [19,20]. Since many stressors may cause both oxidative stress and direct DNA damage, it is vitally important that Dps can both bind to DNA and perform the function of iron oxidation simultaneously and completely independently [21]. Thus, Dps utilizes dual functions during the protection of cells against various adverse stress conditions. At the same time, the protein not only compacts DNA, as other NAPs do, but provides the highest level of structural organization of the nucleoid, the crystalline protein–DNA complex, thereby transforming the dynamic nucleoid into a static, presumably completely inactive structure [1,22]. However, new data have recently been published demonstrating that the transcriptional activity of DNA is partially preserved even during the formation of a protective crystalline complex with Dps [23].

The discovery of crystal structures in living cells and the association of this phenomenon with bacterial drug resistance have generated great interest in the structure of Dps itself and its complex with DNA. Originally, Dps was discovered in *E. coli* in 1992, giving the name to the protein family [19]. The protein structure is well-studied down to the atomic level. Dps is a highly symmetrical dodecameric protein with P23 tetrahedral symmetry assembled from identical subunits with molecular mass of 18.7 kDa each, and with an external macromolecule size of ~9 nm and a central cavity diameter of ~4.5 nm [24]. It is assumed that Dps binds DNA via electrostatic bonds of its lysine-rich N-terminal domains with the negatively charged sugar-phosphate DNA backbone [19,25,26]. However, structural studies of Dps–DNA co-crystals in vitro proved less effective. Attempts to detect and locate the position of DNA in these co-crystals have failed; thus, only speculative models of the Dps–DNA complex structure have been proposed [13,16,17,18,19].

For the first time, the structural organization and lattice parameters of co-crystals of the Dps–DNA complex with a resolution of 13.5 Å have been described in detail quite recently as a result of the application of two complementary methods: small-angle X-ray scattering (SAXS) and cryo-electron tomography (cryo-ET). It was shown that co-crystals with cubic and triclinic unit cells can be formed depending on the buffer parameters and local ion concentration [27,28]. Further studies demonstrated that Dps can co-crystallize with DNA of different lengths (from ~3000 to ~10,000 base pairs) and nucleotide sequences, with one Dps molecule contacting a ~6 nm long DNA segment [29].

Studies directly concerning the biocrystallization processes in living bacterial cells are much more complicated. Crowding and the presence of many different cytoplasmic components, the ratio of which varies greatly depending on external conditions, make it difficult to interpret the results. Among the works devoted to the structural studies of this phenomenon in vivo, the works of A. Minsky with colleagues are of particular importance [30,31,32]. This group of researchers managed to obtain unique information, including demonstration of the appearance of toroidal structures at certain stages of the bacterial stationary phase [32]. The dense circular areas resemble the toroids observed after 24 h of bacterial cell starvation, but do not reveal an ordered organization. It should be noted that such toroidal structures have been repeatedly detected for DNA in vitro under the influence of various condensing agents, and they were suggested to represent a fundamental morphology selected by nature for high-density DNA structures in metabolically quiescent systems [33].

Before investigating such complex phenomena in vivo, a detailed in vitro study is necessary. One of the important conditions for the formation of the complex may be the nature of the DNA itself. So far, we have used DNA fragments of different lengths (from ~3000 to ~10,000 base pairs). The aim of this study is to determine the spatial organization of the *E. coli* genomic DNA in Dps–DNA co-crystals in various buffers, thus continuing the series of studies focused on identifying the features of Dps–DNA complex formation under various conditions in vitro. This study demonstrates different types of genomic DNA and Dps packaging depending on the buffer composition and, for the first time, clear cylindrical structures were observed for these objects in vitro. These new findings are another step toward understanding the structural features of genomic DNA condensation and the associated defense mechanisms of bacterial cells.

## 2. Results

### 2.1. Conditions for the Formation of Dps Complexes with Genomic DNA

In the present work, we studied the formation of Dps complexes with genomic DNA in two fundamentally different buffers: one containing KCl and MgCl_2_ (20 mM Tris-HCl, pH 7.5, 50 mM KCl, 2 mM MgCl_2_, hereinafter Sample 1) and another one containing NaCl and EDTA (20 mM Tris-HCl, pH 8, 50 mM NaCl, 0.5 mM EDTA, hereinafter Sample 2). This choice of buffer compositions was motivated by the following reasons: Firstly, it is well-known that Dps binds DNA non-specifically via electrostatic bonds of its lysine-rich N-terminal domains with the negatively charged sugar-phosphate DNA backbone [19,25,26]. We have previously shown that Mg^2+^ ions block the N-terminals of the Dps protein, preventing its interaction with DNA [34]. Therefore, the interplay between the protein and DNA is largely determined by the salt composition, the ionic strength of the buffer, and the presence of metal divalent ions [34,35,36]. Secondly, this work represents a further step in the study of the phenomenon of protective crystalline Dps–DNA complex formation, and at this stage it should be noted that the formation of such a complex in vivo is preceded by several preparatory steps involving other nucleoid-associated proteins. Among them, integration host factors (IHFs) are some of the most abundant NAPs in *E. coli* at the stationary phase of the bacterial life cycle immediately before Dps–DNA co-crystallization. It was revealed that Dps and IHF are differentially selected for DNA binding, when changing such environmental factors as pH, KCl or MgCl_2_ concentration, and temperature over physiological ranges [37]. As pH and MgCl_2_ concentrations increase, a switch from Dps binding to IHF binding occurs, resulting in a different structural organization of the DNA. This means that the nucleoid can return to a transcriptionally active state when the external conditions are changed. Therefore, the question arises whether the formation of a Dps–DNA complex is possible in a buffer containing potassium and magnesium within the given range and, if so, what the structure of the formed complex is. These issues are important for understanding the processes of crystallization in living cells and, thus, determine the use of the buffer containing KCl and MgCl_2_ in our study.

The buffer in Sample 2 has previously been shown to provide optimal conditions for the formation of Dps–DNA complexes in vitro [28]. In the present work, it is used for the first time as a medium for the formation of the Dps complex with genomic DNA.

### 2.2. Elongated Dps–DNA Biocrystals, Filament-like Assemblies and Their Basic Structural Element

The primary cryo-EM data of Sample 1 show the formation of differently oriented elongated crystals (Figure 1) with a length of several tens of microns and a width up to 500 nm. 

A more detailed study using cryo-ET (Figure 2) revealed that the observed crystals have a triclinic unit cell (Figure 2E,F), with parameters *a* = 9.5 ± 0.4 nm, *b* = 9 ± 0.4 nm, *c* = 9.2 ± 0.4 nm, α = 78 ± 1°, β = 85 ± 1°, γ = 62 ± 1°. The crystals represent a multilayer structure, with each layer consisting of Dps dodecamers with pseudohexagonal packing. This structure corresponds well to Dps packing in the complexes with plasmid DNA (*a* = 9.1 nm, *b* = 9.5 nm, *c* = 10.4 nm, α = 75°, β = 88°, γ = 60°) described in [28] with slight differences in the angle α and vector *c* length (Figure 2F). The decrease in vector *c* length resulted in a reduced spacing between the layers from 10 nm [28] to 9 nm. The lengths of the vectors *b* and *c* differ slightly and virtually correspond to the diameter of the Dps dodecamer, so that the Dps molecules contact their neighbors not only within the pseudohexagonal layers but also between the layers, as can be seen in Figure 2F. The aspect ratio of the observed crystals along the vector *b* (along which parallel DNA strands were oriented in [28]) is significantly larger than that of previously studied Dps crystals with plasmid DNA [27,28].

It is worth noting that individual DNA strands and Dps–DNA complex filament-like assemblies were also observed in the represented sample. These filaments with a thickness of one to three Dps dodecamers lined up along two to four parallelly arranged DNA molecules (Figure 3) and can be considered as nucleation regions of Dps–DNA co-crystals.

Figure 3 shows tomographic slices with Dps particles on small segments of filament-like assemblies of various thickness, demonstrating the spatial organization of DNA and Dps particles. Dps dodecamers are randomly attached to a single DNA strand (Figure 3A). Individual Dps molecules can bind two separate DNA strands (Figure 3B), but the presence of four DNA strands (Figure 3C) leads to the formation of ordered rows of Dps particles (virtually a one-dimensional crystal or filament). Tomographic slices at different heights in Figure A1B–D (Appendix B) show the presence of four DNA strands. Slices of subtomogram averages given different symmetries (Figure A2, Appendix B) show that the density from the DNA strands is most pronounced when C4 symmetry is applied.

Subtomographic averaging (Figure 4) shows that individual rows of Dps dodecamers are surrounded by four parallel DNA strands located at the same distance (about 8.5 nm) from each other (Figure 4A).

Previously, in our study [28], the spatial organization of DNA and Dps was detected in co-crystals, consisting of a set of Dps layers with a dense pseudohexagonal package alternating with layers of parallel DNA strands. Each Dps was surrounded by four strands of DNA, and the crystal was represented as a set of connected rows of adjacent Dps dodecamers (similar to Figure 3C). Thus, a single row of Dps and four mutually parallel DNA strands can be considered as the basic structural element of Dps–DNA co-crystals.

Figure 3D shows two rows of Dps with six DNA strands. It can be clearly seen that the shift between the rows is not equal to half a period and, on average, is ~3 nm, which corresponds to the packing of two rows from different layers in triclinic co-crystals.

A further increase in the number of Dps rows separated by DNA strands is shown in Figure 3E. Nine rows form three layers of Dps (three rows wide) shifted relative to each other by ~3 nm, which provides an average angle *α* ~ 73°. Further growth leads to the formation of well-ordered crystals with triclinic packing, as shown in Figure 2. Nevertheless, an increase in ensemble size does not automatically translate into an increase in order. Figure A3 (see Appendix B) shows a bundle comparable in size to the crystal in Figure 2, albeit lacking long-range order and exhibiting vacancies and variations in Dps particles shifting relative to each other along the DNA axis. However, the bundle represents a set of parallel strands of DNA alternating with Dps rows; on average, the Dps dodecamers from adjacent rows are shifted by half the period of the row.

### 2.3. Cylindrical Assemblies

Examination of the Dps complex with DNA in another buffer (20 mM Tris-HCl, pH 8, 50 mM NaCl, 0.5 mM EDTA, sample 2) indicated the presence of massive bundles with a length of several tens of μm and a width of up to 10 μm (Figure 5A). These complexes include smaller structures—cylindrical assemblies—ranging in length from 0.3 to 4 μm (average: 1.23 μm) and, in width, from 150 to 250 nm (average: 205 nm; Figure 5B,C), which are oriented predominantly along the direction of the bundles. The obtained tomographic slices (Figure 5D,E) demonstrate that they have the shape of half-tubes in the cross section.

Figure 6 and Appendix A illustrate placed-back Dps particles on a small segment of cylindrical assembly. The obtained 3D reconstructions demonstrate that the crystals consist of densely packed 2D sheets (Figure 6A,B), with shifts between the Dps layers (Figure 6B) that can be observed in the crystal section (Figure 6D). The majority of the observed half-tubes consist of four to eight Dps layers with a curvature radius ranging from 40 to 130 nm. The different curvature radii preclude translation between the protein layers, as demonstrated in Figure 5D.

Figure 7 represents the results of subtomogram averaging and the corresponding slices of averaged subtomograms demonstrating the mutual arrangement of Dps and genomic DNA in Sample 2. Similar to Sample 1, the alternation of the Dps and DNA layers in such complexes is revealed, as was earlier reported in our study [28] for crystals with triclinic unit cells. The distance between the Dps layers is 9.9 nm, and the distance between neighboring Dps particles is ~9 nm.

Figure 7C,D shows a well-resolved density from the DNA strands, while Figure 7E,F clearly displays their curvature. The density for Dps is resolved poorer outside the central densely packed layer, which may be attributed to the lack of a fixed shift between neighboring layers.

It is worth noting that the mutual arrangement of Dps particles and DNA strands in the central layer of cylindrical assemblies corresponds exactly to the position of Dps particles and DNA strands in filament-like assemblies. However, in this case, DNA strands are perpendicular to the direction of the half-tube axis.

According to our estimates, each cylindrical assembly contains 5 × 10^3^ to 6 × 10^4^ dodecamers of Dps (on average about 2 × 10^4^), thus containing 1 × 10^5^ to 1.2 × 10^6^ base pairs of DNA.

In addition to the cylindrical assemblies, Sample 2 also contains bundles of Dps–DNA ensembles partially ordered into co-crystals up to 500 nm in size (see Figure A4, Appendix B). Analysis of the crystal lattice revealed that the bundle consists of smaller co-crystals having triclinic unit cells. Importantly, their orientation is not random with vector *b* (along which the DNA is oriented) in all co-crystals directed along the bundle. For these crystals, a high aspect ratio is not observed.

## 3. Discussion

Three-dimensional reconstructions of all Dps complexes with genomic DNA studied in this work are shown in Figure 8. In both samples, complexes with parallel DNA strands are observed on a small scale. Moreover, in contrast to the complexes with short plasmid DNA, co-crystals with cubic unit cells, wherein DNA strands are arranged in three mutually perpendicular directions (such as reported in our work [27]), were not observed, presumably due to the length of genomic DNA, which prevents such compactization.

Therefore, in the case of genomic DNA, the main structural element of all observed assemblies (Figure 8) is a one-dimensional row of contacting Dps particles surrounded by four parallel strands of DNA. This element is present in the filament-like structure (Figure 8A), in co-crystals with triclinic unit cells (Figure 8B), and in the cylindrical shape (Figure 8C). 

Despite the fact that Sample 1 contains divalent magnesium ions, the formation of the Dps–DNA crystalline complex still occurred contrary to our earlier data [34]. However, the fact is that there are threshold values of Mg^2+^ concentrations required for the formation of Dps–DNA complex, which were determined in [26,37,38]. It was found that the complex is formed at concentrations of MgCl_2_ less than 2 mM, while higher concentrations of the cation (7.5–10 mM) inhibit Dps–DNA complex formation. In a previous work [34], a concentration of 20 mM MgCl_2_ was used, while in the present study, a threshold value of 2 mM MgCl_2_ was applied.

The formation of parallel DNA strands, alternating with DNA binding proteins, is not unique for Dps. Similarly, HUα proteins can bind the nucleoid, as shown by X-ray diffraction and SAXS [39]. In this case, the DNA is not subject to the significant folding typical of histones and other NAPs. Moreover, different strands of DNA are connected by HUα as bridges, and HUα–DNA complexes may contain a gap between HUα proteins, which can be used by other proteins for DNA transcription. It should be noted here that although the nucleoid is condensed maximally, forming a crystalline complex with Dps during starvation or exposure to stressors, it has also been shown that Dps does not completely suppress transcription. A previous publication [23] suggests that rather than forming static crystalline structures, Dps forms dynamic complexes with similar diffusive properties to liquid–liquid phase-separated organelles. This dynamic behavior indicates that Dps complexes may retain some features of a fluid. In addition, it was demonstrated that Dps blocks restriction endonucleases, but not RNA polymerases, from binding to DNA, while retaining the possibility of transcriptional activity [23].

It is also interesting to note that both high- (Figure 2) and low-ordering (Figure A3, Appendix B) Dps–DNA ensembles observed in this study are of comparable size, suggesting that Dps–DNA co-crystallization is not described by classical nucleation theory [40]. A similar effect has been observed for glucose isomerase, which assembles into rows called nanorods—basic structural units of the self-assembly process, which leads to the formation of oligomers and fiber-like assemblies during the initial stages of crystallization. Studies of protein crystallization of glucose isomerase have shown that assembly size and crystallinity are order parameters that can evolve independently [41]. 

The important result of the present work is the detection of cylindrical structures in Sample 2. To the best of our knowledge, such structures have not been shown before. However, when studying the compactization of DNA, toroidal structures, i.e., structures resembling a spirally wound rope, are most often observed. Toroidal shapes of DNA have received a great deal of attention because of their relationship to gene packaging in some viruses, and also because of their potential use in gene therapy. DNA is a natural semi-flexible polyelectrolyte that is considered to be an excellent model system for studying the formation of nanoparticles during the collapse of charged polymers. In a published review [42], DNA toroids are defined as condensates with an approximately circular cross section that are typically formed from dilute solutions of DNA upon the addition of multivalent cations. However, this definition cannot be attributed to the cylindrical structures observed in the present work, which were formed during the interaction of DNA with Dps proteins in the absence of divalent magnesium ions. 

In one of the most important works concerning the formation of toroidal structures in living *E. coli* cells under starvation [32], the authors concluded that the main factor of toroid formation in vivo is the Dps protein, but the toroidal shape of chromatin is characteristic only for the 24 h starvation period of *E. coli* bacteria. Further, during starvation, toroidal morphology, acting as a structural template, promotes the formation of hexagonal crystal lattices of Dps–DNA through epitaxial growth, and after 48 h of starvation, the toroidal structures completely disappear. The authors also proposed a hypothetical model of this intermediate assembly, where the DNA is localized between the pseudohexagonal faces of the Dps dodecameric particles. However, the distance between alternating layers of the presumed Dps–DNA assemblies was estimated at 6.8–7.5 nm, leaving insufficient space for the 9 nm Dps proteins in these layers in the presence of ~2.4 nm thick DNA strands, even with a possible ~20% shrinkage under an electron beam. Therefore, it can be assumed that the nature of the formation of toroidal structures might not be associated with Dps, but rather with other intracellular factors, among which some NAPs are the most likely candidates. We would also like to note that the model proposed in [32] is more similar to the cylindrical structures demonstrated in the present work than to toroids.

Based on the results of our previous works [27,28] and findings of the present investigation, we propose a hypothesis that somewhat differently represents the process of formation of toroidal DNA–Dps complexes during cell starvation. We suggest that the nature of the formation of transient toroidal structures may not be associated with Dps, but rather with other intracellular factors. One of the most likely candidates is the IHF protein, which dominates in the late stationary phase just before Dps–DNA co-crystallization. The small size of this protein allows it to fit between parallel strands of DNA, forming a toroidal structure with an interplanar spacing of 6.8–7.5 nm, as was determined in [32]. In the late stationary phase, IHF molecules are replaced by Dps, which, even in the absence of DNA, forms a multilayer structure with hexagonal dodecamer packing [43] and, thus, determines the structure of the resulting DNA–Dps crystalline complex. This replacement is possible because it was revealed earlier that Dps and IHF are differentially selected for DNA binding, when changing environmental factors [37]. 

Another important representative of NAPs, HU proteins, are also present during the late period of bacterial cell starvation. HU and Dps together form complex multiphasic droplets with DNA, without undergoing complete mixing, with evidence of the formation of domains of differential HU or Dps content. This ability of HU condensates to accommodate Dps-rich condensates could prevent the premature crystallization of Dps during stress [44]. HU and IHF are close homologues with similar structures and functions. Therefore, it can be assumed that both proteins can participate in the formation of toroidal shapes during late periods of cell starvation. To determine their exact role, special biological experiments with knockout bacteria are needed, in which the genes encoding one or another of the studied proteins will be sequentially deleted. These experiments are yet to be carried out.

However, at this stage, it can be assumed that the initial protein–DNA toroids in starved *E. coli* cells represent a transient form, which acts as a matrix for subsequent formation of Dps–DNA co-crystals. In the present work, cylindrical structures, previously unknown, were obtained for the first time using Dps proteins and *E. coli* genomic DNA in various buffers in vitro.

## 4. Materials and Methods

### 4.1. Dps and DNA Isolation and Purification

Highly purified Dps proteins were obtained in a prokaryotic expression system according to an earlier protocol [27,28].

*E. coli* genomic DNA samples were produced and purified according to the protocol [45,46]. Quality control of the isolated DNA was performed using electrophoresis in an agarose gel, where DNA fragments were arranged at a high molecular weight, approximately 25 kbp.

One buffer variant for genomic DNA compared with the plasmid DNA buffer used previously [27,28] additionally contains KCl and MgCl_2_ in physiological ranges. The concentration of MgCl_2_ is significantly lower than that previously used [34,35] since, according to the paper [37], this type of buffer is closer to the native conditions of the cell cytoplasm. 

Thus, in the present work, we studied (for the first time) Dps complexes with genomic DNA in a 1:5 weight ratio in two buffer variants: 20 mM Tris-HCl, pH 7.5, 50 mM KCl, 2 mM MgCl_2_ (Sample 1); 20 mM Tris-HCl, pH 8, 50 mM NaCl, 0.5 mM EDTA (Sample 2). 

### 4.2. Preparation for the Cryo-EM Study

The sample solution mixture of Dps and genomic DNA was incubated for at least 10 min after mixing at a temperature of 20 °C. A total of 3 µL of the sample solution mixture was applied to a Lacey EM grid (Ted Pella, USA) glow discharged for 30 s at 0.26 mbar of pressure using a current of 25 mA with Pelco EasiGlow (Ted Pella, Northport, NY, USA). The grids were then blotted with filter paper for 2.5 s on both sides at T = 20 °C and 95–100% humidity and vitrified using Vitrobot Mark IV (Thermo Fisher Scientific, Hillsboro, OR, USA). Frozen grids were transferred to cryo-TEM in liquid nitrogen.

Preliminary studies of vitrified samples were performed using a Tecnai G2 SPIRIT cryo-TEM (Thermo Fisher Scientific, USA) equipped with an Eagle CCD detector (Thermo Fisher Scientific, USA) with a resolution of 4096 × 4096 pixels. The studies were performed at an accelerating voltage of 120 keV in the low-dose mode.

### 4.3. Cryo-Electron Tomography Study

Cryo-ET was performed using a Titan Krios 60–300 cryo-TEM (Thermo Fisher Scientific, USA) equipped with Falcon II direct electron detector (Thermo Fisher Scientific, USA) and Cs corrector (CEOS, Heidelberg, Germany) at 300 kV. Experimental data were obtained using tomography software (Thermo Fisher Scientific, USA) at a magnification of 18,000× (pixel size 3.7 Å), exposure of 2 s, and defocus in the range [−4.0;−6.0] μm. The total electron dose was 100 e/Å^2^.

Cryo-ET data were processed using IMOD [47]. Tomograms were filtered using IsoNet [48] to display tomographic slices and identify particles. Particle coordinates were determined automatically using EMAN2 [49,50] (similar to the procedure described in [27]). Subtomogram averaging was performed using Relion 4.0 [51,52]. For the final averaging, 717 subtomograms were used for the cylindrical complexes, 52 subtomograms (manually selected) for the filament-like assemblies, and 659 subtomograms for nanocrystals with triclinic unit cells. The UCSF Chimera [53] software package was used for imaging; FFTs were performed with ImageJ [54].

## 5. Conclusions

Studying the processes of stress-induced biocrystallization directly in living bacterial cells is extremely difficult due to the large number (crowding) of cell components and their continuous interaction with each other and with DNA in response to environmental changes. A necessary alternative is to study the structure and function of individual cellular elements and/or their interactions in vitro, i.e., to imitate vital cell responses to stress by studying selected processes of particular importance. This work represents the next step in our sequential structural investigation of the protective crystalline complex of DNA with Dps proteins, which is produced at the late stage of exponential cell growth. The use of two different buffer systems allowed us to simulate the response of the cell nucleoid to changes in such environmental parameters as salt composition and the presence or absence of certain metal cations over physiological ranges. In this study, for the first time, we used genomic DNA, in contrast to our previous works, where we explored the interactions of Dps with DNA of a certain length (from 3000 to 10,000 base pairs).

We have demonstrated various morphologies of the complexes: fibrous structures, “half-tubes”, and co-crystals with triclinic unit cells. A common feature of these various formations was the presence of a structural group, consisting of a single row of Dps and four mutually parallel DNA strands. Such a group can be considered as the basic structural element of Dps–DNA co-crystals. 

The most interesting result was the detection of cylindrical structures within the Dps–DNA complex formed in a buffer that does not contain divalent magnesium cations. These structures resemble the hypothetical model of the intracellular DNA–Dps assembly proposed in [32]. However, analysis of the literature and available experimental data allows us to suggest that instead of the large Dps protein, smaller NAPs—such as IHF and HU, which are actively expressed at an early stage of cell starvation—might be involved.

Overall, the present work continues the series of studies on the morphological diversity of Dps–DNA complexes under various in vitro conditions, which contribute to a deeper and more detailed understanding of the processes occurring in living cells. In the course of this research, for the first time, the formation of cylindrical structures in vitro has been shown and described in detail. The next logical step involves the in vitro study of Dps–DNA complex formation in the presence of various intracellular agents, including some NAPs.

## Figures and Tables

**Figure 1 ijms-24-08534-f001:**
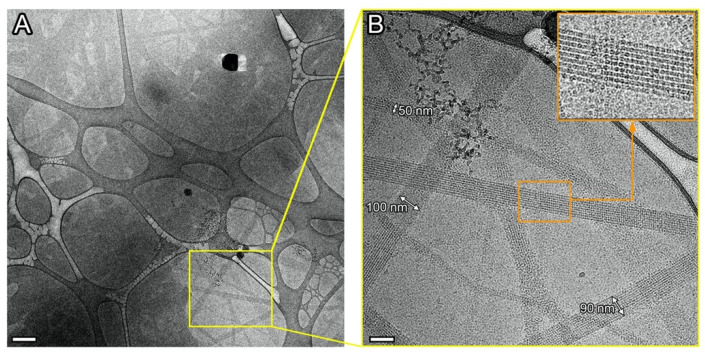
Cryo-EM images of elongated crystals in Sample 1. (**A**) An overview of the sample and (**B**) enlarged image of the marked area. Scale bars are 500 nm (**A**) and 100 nm (**B**).

**Figure 2 ijms-24-08534-f002:**
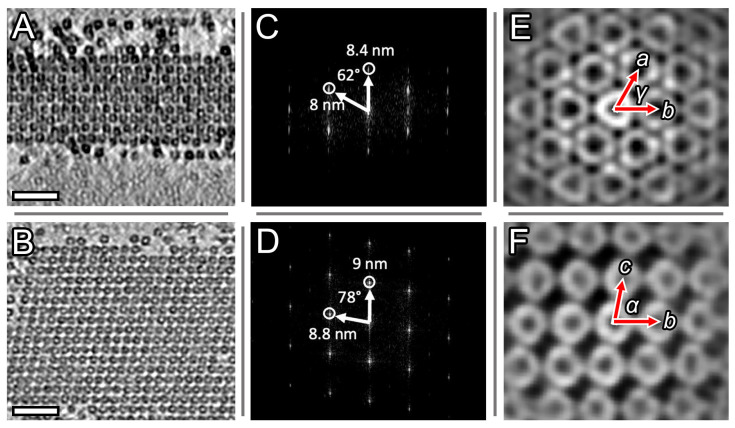
Tomographic slices of biocrystals in Sample 1. (**A**) slice with the XZ plane; (**B**) slice with the XY plane; (**C**,**D**) corresponding Fast Fourier Transforms (FFTs). (**E**,**F**) corresponding slices of the averaged subtomogram with marked vectors and angles of the triclinic cell. Scale bars (**A**,**B**) are 50 nm.

**Figure 3 ijms-24-08534-f003:**
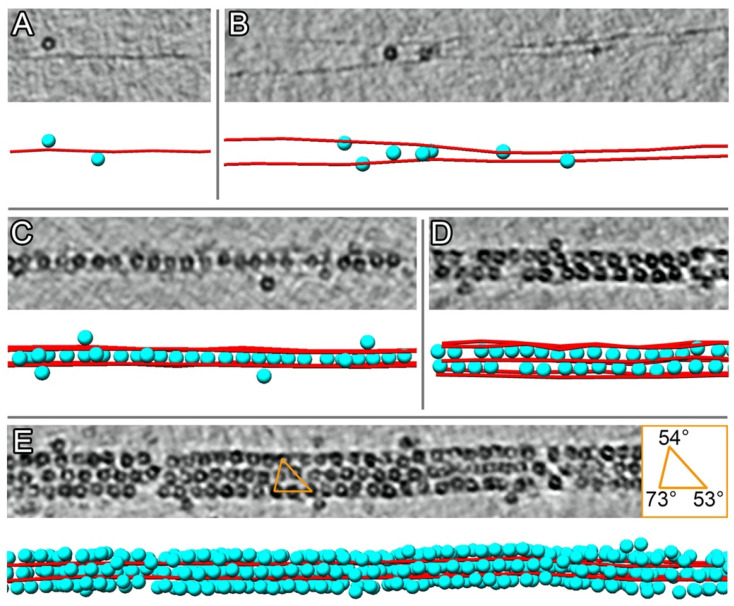
Tomographic slices (XY plane) and corresponding 3D reconstructions (placed-back subtomogram averages) of filament-like segments in Sample 1. (**A**) single DNA and Dps particles; (**B**) two DNA strands bridged by Dps particles; (**C**–**E**) weakly ordered complexes consisting of one, two, and nine Dps rows alternating with DNA strands. The reconstructions show Dps particles in cyan and DNA in red.

**Figure 4 ijms-24-08534-f004:**
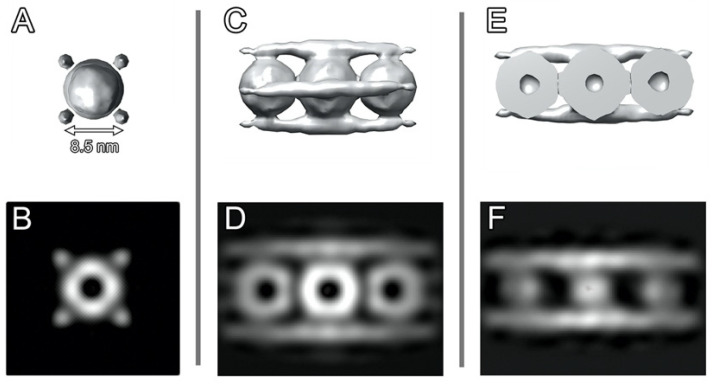
Subtomogram averaging result of the filament-like Dps–DNA assembly: isosurfaces (**A**,**C**,**E**) and slices (**B**,**D**,**F**) of averaged subtomograms showing the mutual arrangement of Dps and genomic DNA.

**Figure 5 ijms-24-08534-f005:**
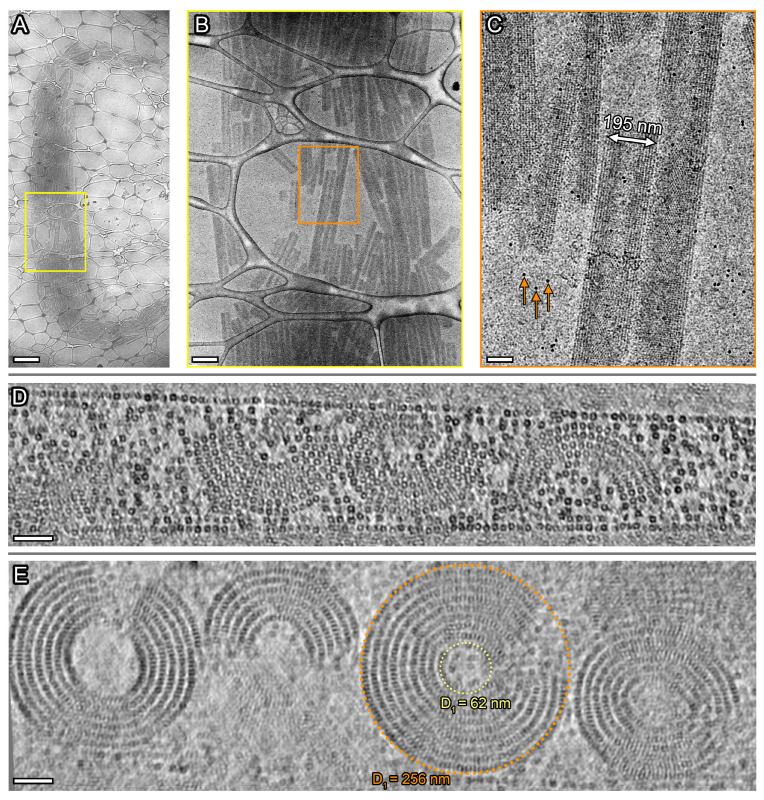
Cryo-EM images of biocrystals (**A**–**C**) and tomographic slices of the biocrystal cross section ((**D**,**E**); XZ plane) in Sample 2. The scale bars are 2 μm (**A**), 500 nm (**B**), 100 nm (**C**), 50 nm (**D**,**E**). The slice thickness is ~1.1 nm (**D**) and ~22 nm (**E**). Arrows (**C**) indicate the colloidal gold nanoparticles added to the sample prior to cryo-ET experiments.

**Figure 6 ijms-24-08534-f006:**
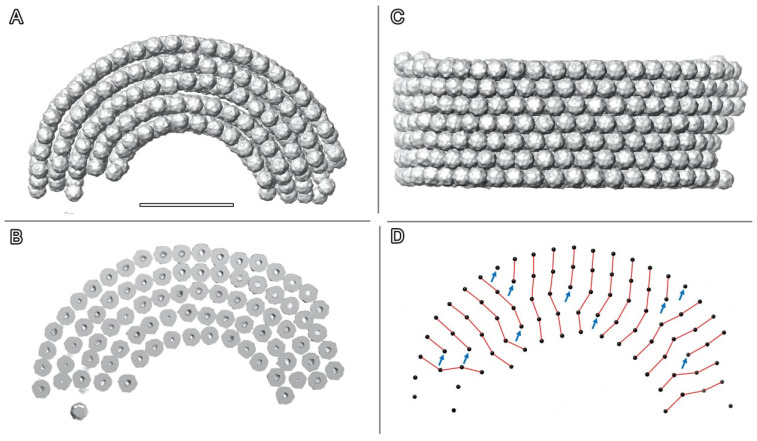
3D reconstruction of the Dps particles in the observed crystals. (**A**) view across the half-tube axis, (**B**) section across the half-tube axis, (**C**) view along the half-tube axis; (**D**) packing pattern across the half-tube axis. Black dots indicate positions of Dps centers; red lines indicate nearest neighbors of Dps in different layers; blue arrows indicate Dps particles without a match in the previous layer. The scale bar is 50 nm.

**Figure 7 ijms-24-08534-f007:**
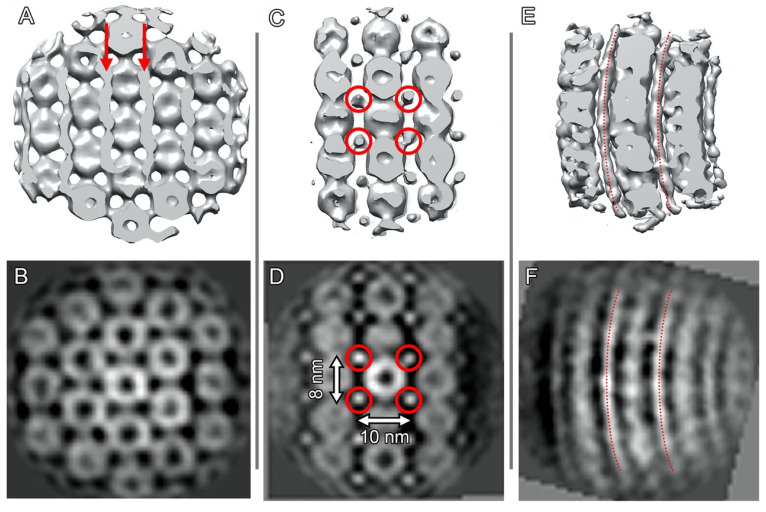
Subtomogram averaging results showing the mutual arrangement of Dps and genomic DNA in Sample 2: isosurfaces (**A**,**C**,**E**) and corresponding slices of averaged subtomograms (**B**,**D**,**F**) in different orientations. Red arrows and circles indicate DNA strands in different orientations.

**Figure 8 ijms-24-08534-f008:**
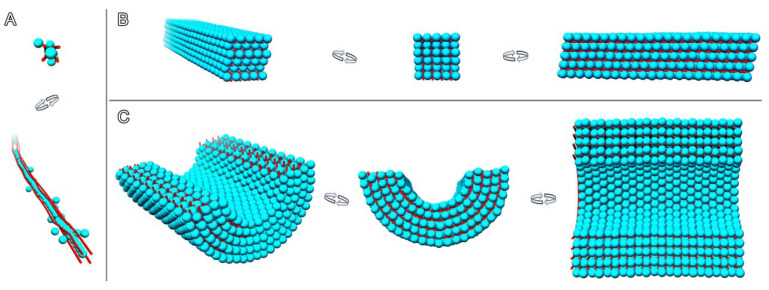
Different types of packing of the Dps–DNA complex. (**A**) filament-like assembly; (**B**) nanocrystal with triclinic unit cell; (**C**) cylindrical assembly. (**A**,**B**) Sample 1, (**C**) Sample 2. Dps particles are shown in cyan, DNA in red.

## Data Availability

All data are available in the manuscript. Electron microscopy density maps are deposited in the EMDB (EMD-16439—Dps–DNA filament-like assembly; EMD-16438—cylindrical Dps–DNA assembly).

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
