# Peer review of "Morphological Diversity of Dps Complex with Genomic DNA"

_ijms, 2023, doi:10.3390/ijms24108534_

Round 1

Reviewer 1 Report

This research paper shows very unique results.
Research using cryo-EM at 2022 year after what was shown in the authors' paper, AFM in 21, shows a distinct difference in Dps and DNA binding.
Contrary to the results of the authors in the previous 2022 year, this structural study using E. coli's genome DNA is impressive to form a regular Dps structure that forms the cylindrical structure shown in sample 2 with EDTA added. It is believed that Dps has revealed the answer to its specific action and binding since it is known to perform the function of protecting DNA. Of course, the action of Dps still acting under in vivo conditions may differ from the results of this study because it will be affected by various factors in E. coli, especially NAPs, and ion concentrations. Nevertheless, the results of this study will provide researchers in this field with quite fresh ideas. The results of this study are suitable for publication, and it would be better to correct some of the minor problems below.
- The arrow marks in Figure 5 are shown in Figure 5c.

Reviewer 2 Report

The manuscript by Chesnokov et al reports the study of the formation of complexes (biocrystals) of E. coli Dps with DNA. The work has the novelty of using genomic DNA. It presents a set of solid and convincing data, which adds to the state of knowledge of this very interesting group of proteins. I do, however, have some questions and therefore suggest minor revisions.

1)     Introduction section - It would be interesting if the authors included a section on the regulation of the two main functions of the protein. In biocrystals does the protein maintain its iron oxidation and mineralization activity?

2)     Lines 43 and 44 – The authors state that “… the crystalline protein-DNA complex, thereby transforming 43 the dynamic nucleoid into a static, predominantly inactive structure”, but later (lines 301-304) claim that transcription is not fully impaired and that the complexes are dynamic. Please clarify.

3)     Line 122 – Please include a reference on the effect of the ionic strength on the tertiary and quaternary structures of the protein under study. 

4)     About the buffers used to test the formation of Dps-DNA complexes: how is the structure of the protein affected? Why the pH change in both buffers? What is the role of the EDTA present in buffer 2? What was the rationale for using KCl in buffer 1, while using NaCl in buffer 2?

5)     Lines 181-183. How are the Dps molecules arranged in these filaments, precursors of biocrystals? Given that they are weakly ordered, does that imply that during nucleation and formation of the biocrystal there are rearrangements? Could the authors propose a mechanism?

6)     Lines 257-258 – Please use scientific notation.

7)     About Figure 8 – What is the relative percentage of each structure? It has been reported that compaction of DNA occurs via protein aggregation, but it seems not to be the case in these complexes. Could the authors discuss this issue?

8)     How dynamic and/or reversible are these different DNA complexes? In case they are, what triggers these processes? Is the protein in these complexes catalytically active?

9)     Lines 289-292 – Please elaborate; one assumes that the samples are homogeneous. For instance, the formation of the filaments is favored at higher or lower MgCl2 concentrations?

10)  Line 377 – The authors claim “… contains KCl and MgCl2 in physiological ranges”. Please specify the physiological intracellular concentrations of the K+, Cl and Mg2+

11)  Line 381 – Please clarify the Dps/genomic DNA molar ratio used in the DNA binding reaction – 1:5, meaning that the DNA is in a 5x excess?

12)  Line 385 – Please indicate the conditions of the DNA binding reaction, namely temperature and reaction time.

13)  Please correct the legends of Figure 7 (B,D,F instead of B,D,E) and Figure A4 (F, G instead of E, G).

Reviewer 3 Report

The manuscript entitled “Morphological Diversity of Dps Complex with Genomic DNA” from Yuri Chesnokov et al is interesting. However, the manuscript should be revised with following suggestions/comments for enabling it towards publication. The structure of the thesis needs to be optimised and some of the experimental results may need to be supplemented.

1. The authors are advised to rewrite the abstract. The abstract section needs to outline not only the background of the paper, but also a summary description of the experimental results and conclusions.

2. The third and fourth paragraphs of the Introduction are lengthy. In the third paragraph, it is suggested that the authors summarise the previous research on Dps-DNA complex rather than listing it all. The fourth paragraph also needs to be streamlined.

3. It is strongly recommended that the authors explain the structure of the Dps Complex with Genomic DNA in this paper by means of molecular calculations, for example using HADDOCK or referring 10.1093/nar/gkz256.

4. This article contains some spelling errors and subheading numbering errors that need to be corrected by the author.

5. In the Results and Discussion section, the author needs to subdivide the content and add appropriate subheadings. A direct statement of the results of the experiment is not conducive to the reader understanding the scientific ideas of the paper. Do not just present the experimental phenomena for each figure.

6. The authors are asked to consider whether the experimental results of Figure 3A and 3B need to be shown, as if the results of Figure3C-3E only need to be discussed.

7. It is recommended that the authors' INTRODUCTION section mentions some of the common NAPs and analyses the biological significance of biocrystallization.

The manuscript entitled “Morphological Diversity of Dps Complex with Genomic DNA” from Yuri Chesnokov et al is interesting. However, the manuscript should be revised with following suggestions/comments for enabling it towards publication. The structure of the thesis needs to be optimised and some of the experimental results may need to be supplemented.

1. The authors are advised to rewrite the abstract. The abstract section needs to outline not only the background of the paper, but also a summary description of the experimental results and conclusions.

2. The third and fourth paragraphs of the Introduction are lengthy. In the third paragraph, it is suggested that the authors summarise the previous research on Dps-DNA complex rather than listing it all. The fourth paragraph also needs to be streamlined.

3. It is strongly recommended that the authors explain the structure of the Dps Complex with Genomic DNA in this paper by means of molecular calculations, for example using HADDOCK or referring 10.1093/nar/gkz256.

4. This article contains some spelling errors and subheading numbering errors that need to be corrected by the author.

5. In the Results and Discussion section, the author needs to subdivide the content and add appropriate subheadings. A direct statement of the results of the experiment is not conducive to the reader understanding the scientific ideas of the paper. Do not just present the experimental phenomena for each figure.

6. The authors are asked to consider whether the experimental results of Figure 3A and 3B need to be shown, as if the results of Figure3C-3E only need to be discussed.

7. It is recommended that the authors' INTRODUCTION section mentions some of the common NAPs and analyses the biological significance of biocrystallization.
